# Enrichment of Bone Tissue with Antibacterially Effective Amounts of Nitric Oxide Derivatives by Treatment with Dielectric Barrier Discharge Plasmas Optimized for Nitrogen Oxide Chemistry

**DOI:** 10.3390/biomedicines11020244

**Published:** 2023-01-17

**Authors:** Dennis Feibel, Alexander Kwiatkowski, Christian Opländer, Gerrit Grieb, Joachim Windolf, Christoph V. Suschek

**Affiliations:** 1Department for Orthopedics and Trauma Surgery, Medical Faculty, Heinrich-Heine-University Düsseldorf, Moorenstraße 5, 40225 Düsseldorf, Germany; 2Institute for Research in Operative Medicine (IFOM), Cologne-Merheim Medical Center, University Witten/Herdecke, 58455 Witten-Herdecke, Germany; 3Department of Plastic Surgery and Hand Surgery, Burn Centre, Medical Faculty, RWTH Aachen University, 52074 Aachen, Germany

**Keywords:** dielectric barrier discharge (DBD), cold atmospheric plasma (CAP), nitric oxide radical (NO)

## Abstract

Cold atmospheric plasmas (CAPs) generated by dielectric barrier discharge (DBD), particularly those containing higher amounts of nitric oxide (NO) or NO derivates (NOD), are attracting increasing interest in medical fields. In the present study, we, for the first time, evaluated DBD-CAP-induced NOD accumulation and therapeutically relevant NO release in calcified bone tissue. This knowledge is of great importance for the development of new therapies against bacterial-infectious complications during bone healing, such as osteitis or osteomyelitis. We found that by modulating the power dissipation in the discharge, it is possible (1) to significantly increase the uptake of NODs in bone tissue, even into deeper regions, (2) to significantly decrease the pH in CAP-exposed bone tissue, (3) to induce a long-lasting and modulable NO production in the bone samples as well as (4) to significantly protect the treated bone tissue against bacterial contaminations, and to induce a strong bactericidal effect in bacterially infected bone samples. Our results strongly suggest that the current DBD technology opens up effective NO-based therapy options in the treatment of local bacterial infections of the bone tissue through the possibility of a targeted modulation of the NOD content in the generated CAPs.

## 1. Introduction

Cold atmospheric plasmas (CAP) generated with the help of modern plasma technology under atmospheric pressure conditions allow for interaction with living biological tissues with few side effects due to the low process temperatures [1]. This results in manifold promising therapy options for the treatment of various diseases in humans and animals [2,3]. With the CAPs, a distinction is made between “direct” and “indirect” plasma [4]. Put simply, an indirect plasma is created by ionizing a gas that flows through an electric field. The resulting outflowing plasma can then be applied to the area to be treated in the form of a “plasma jet”. Direct plasma can be generated with the help of dielectric barrier discharge (DBD) technology, with the sample to be treated serving as a counter electrode to the DBD device electrode. In the gap between the electrode of the DBD source and the object to be treated, the ambient atmosphere is ionized, and the generated CAP directly exerts its effect on the treated tissue area [4].

CAPs contain a mixture of different oxygen, nitric oxide and other radical species, UV radiation and a strong flow of charges. All components individually or in combination have the potential ability to influence biological functions if they are applied directly to tissue or cells [1]. Plasma generated with ambient air includes reactive nitrogen species (RNS) such as nitric oxide (NO), nitrogen dioxide (NO_2_) and reactive oxygen species (ROS) such as hydrogen peroxide (H_2_O_2_), ozone (O_3_), superoxide radicals (O_2_^−^), hydroxyl radicals (OH), and many other products in different concentrations and compositions [5,6]. The content of NO and other NO derivatives (NODs) in DBD-generated CAPs are of particular importance. In the context of tissue homeostasis, NO plays a decisive physiological role, e.g., in the regulation of vasodilation, thrombogenesis, the immune and inflammatory response, angiogenesis, cell proliferation and cell differentiation, collagen metabolism, cell death via apoptosis or necrosis, as well as antibacterial defense [7]. In the human organism, NO can be generated enzymatically from the amino acid L-arginine by at least three isoenzymes from the family of NO synthases in probably every cell type [8]. Due to its pivotal role in the regulation of tissue homeostasis, insufficient physiological local NO production or NO availability correlates, for example, with the well-known clinical picture of chronic, bacterially infected and poorly healing wounds [9]. With regard to the therapy of chronic wounds, the positive effect of NO-based therapies could be demonstrated by exogenously applied NO gas, NO donors or NO-containing plasmas which all exerted significant improvement in the wound healing status, including a clear and strong reduction in the bacterial load on the treated wounds [10,11,12]. Since the bacterial infestation of wounds is apparently a decisive driver for the delay in wound healing, it is not surprising that other NO-independent measures that lead to the reduction in bacterial infestation can also positively support wound healing, such as plasma compositions, which are more characterized by a dominant H_2_O_2_ generation and less NO chemistry [13]. It must be noted, however, that in addition to its pronounced antibacterial effect, NO can also induce numerous other wound healing-supporting mechanisms and thus have additional positive properties that H_2_O_2_ and other ROS do not have.

As part of a previous study to evaluate the therapeutic significance of NO-containing DBD-CAPs, we observed that the treatment of human skin with CAPs, which were generated with a power dissipated in the discharge of 122 mW by the same DBD source as we did for the current study, led to an accumulation of considerable amounts of NO derivatives and a decrease in the pH in the treated skin tissue. [14]. Both parameters led to a release of NO in the skin and the development of physiological NO effects, such as enhanced local vasodilation [15]. Interestingly, such a situation did not lead to the occurrence of increased cell toxicity in the treated skin sample, either in vitro or in vivo [16]. In the treated skin tissue, an NO-releasing system could thus be built up with NO-containing CAPs, in which the duration of the NO release and the level of NO production were dependent on the amount of accumulated NO derivatives and thus could be varied solely by the duration of the plasma treatment [15].

Due to the necessity of a free anatomical access of a plasma source to the surface to be treated, plasma-oriented therapies appear to be predestined for the treatment of diseases from the dermatological or stomatological context. In cases of surgical interventions, however, this technique could be used if an otherwise not freely accessible area was opened as part of a medical procedure, like a revision. This would be the case, for example, in the context of supportive local therapy for bacterial complications in bone tissue, e.g., in the clinical picture of osteitis or osteomyelitis. These are often dramatic infectious and inflammatory complications of bone healing caused by bacteria [17], accompanied by the formation of biofilms, overactivation of immune competent cells and irreversible destruction of the bone substance [18]. In addition to systemic antibiosis, the treatment of these complications often consists of surgical debridement of the focus of infection and local use of antibiotic measures [17,18]. Due to the bactericidal properties of NO against free-living bacteria as well as against bacteria organized in a biofilm [19,20], its anti-inflammatory and immunomodulating effects [21] and positive properties in the regulation of osteogenesis [22,23], the use of NO-containing CAPs could represent an effective therapy option in the treatment of osteitis or osteomyelitis.

To date, there are no meaningful studies on whether and to what extent bone tissue can interact with CAP-induced NO derivatives and what bactericidal potential such an interaction can cause. Therefore, in the present study, we used a DBD plasma device which enabled us to achieve varying degrees of accumulation of NODs in exposed bone tissue. We investigated the depth of penetration of the NODs into the bone tissue, the degree of NO release and the antibacterial properties of the NO-optimized CAPs used in different models of bacterial infection of bone tissue. Our results clearly show that the use of plasma technologies, with the help of which a pharmacologically relevant NO release can be induced in exposed bone tissue, represents an effective supportive therapy option in the treatment of bacterial complications of the bone.

## 2. Materials and Methods

### 2.1. Materials

If not indicated otherwise, all chemicals were obtained from Sigma-Aldrich, Merck KGaA (Darmstadt, Germany).

### 2.2. DBD Plasma Source

We used a prototype DBD device which consists of one driven, cylindrical copper electrode covered with aluminum oxide and with a total diameter of 10 mm, exactly as described and characterized recently [15]. CAP was ignited in the gap (1 mm) between this dielectric covered electrode and grounded samples (water, agar, bone tissue), which represented the counter electrode. We have evaluated the effect of CAPs generated under four different device settings. CAPs were generated in ambient air by applying 13.5 kV or 17 kV maximum voltage pulses (kV) combined with trigger frequencies of 300 or 600 Hz (13.5 kV/300 Hz, 13.5 kV/600 Hz, 17 kV/300 Hz, 17 kV/600 Hz) which, with an air gap of 1 mm and positive polarity, corresponded to the power dissipated in the discharge of 122 mW, 275 mW, 221 mW and 410 mW, respectively.

### 2.3. Determination of pH Values

At the corresponding times of the respective experiments, the pH values were quantified by means of the Calimatic 766 pH meter (Knick, Berlin, Germany), both before and after the treatment of the samples with the respective CAPs. A combination pH electrode with a flat membrane (InLab-surface, Mettler-Toledo, Giessen, Germany) was used to measure the pH on solid objects (agar, bone surfaces).

### 2.4. Detection and Quantification of the NO Derivatives

The concentrations of nitrite and other chemically (iodine) reducible nitric oxide derivates in plasma treated aqueous solutions or aqueous exudates of plasma-exposed bone samples were quantified by an iodine/iodide-based assay, and using the NO-analyzer CLD 88 (Ecophysics) exactly as described previously [24]. Additionally, in order to determine nitrate, samples were incubated with 0.1 mol/l vanadium (III) chloride in 1 M hydrochloric acid refluxing at 95 °C under nitrogen [25]. The NOD quantities were quantified using nitrite/nitrate standards in standardized calibration solutions. The nitrate concentrations were calculated by subtracting the nitrite values obtained from the total NOD values.

To quantify the NO gas release from DBD plasma-treated aqueous solution, the respective solution was transferred to a quartz glass cylinder after the CAP exposure (20 mL) and permanently flushed through with a constant gas flow with N_2_ (100 mL N_2_/min). The resulting N_2_/NO mixture was evacuated at the same flow rate and fed into the CLD for quantitative analysis [24]. The integral calculation of NO release from the solution was carried out using a specially designed program using the Matlab software (The MathWorks, Inc., Natick, MA, USA).

In order to quantify the release of native NO gas from human bone samples treated with DBD plasma, the bone preparations were transferred to an airtight chamber (35.2 cm^3^) immediately after the plasma exposure. The chamber loaded with the bone specimen was flushed with an inert gas (N_2_) for 1 min to remove oxygen. Then the chamber was hermetically sealed using two closing valves and NO emanating from the bone samples was accumulated in the chamber for 120 s. After opening the valves, the contents of the chamber were then fed into the CLD analyzer, and the NO content was determined [24].

### 2.5. Characterization of Accumulation of Nitric Oxide Derivates in CAP-exposed Aqueous Samples

As part of preliminary tests, we evaluated the power dissipated in the discharge at which the greatest accumulation of NO derivatives into the exposed solution could be achieved. To do this, we treated 20 mL of H_2_O_dest_ in 50 mL centrifuge tubes with CAP generated under conditions of the device settings mentioned above for a period of 1, 5, 10 or 15 min. We then quantified the pH of the solution and the CAP-dependent accumulation of iodine-reducible NO derivates into the exposed solution. The time-resolved release of NO from the plasma-treated sample were quantified by using the chemiluminescence based NO-analyzer CLD 88 as mentioned above.

### 2.6. Bone Samples

We used pre-machined bovine bone chips (IDS, Boldon, UK), consisting exclusively of cortical tissue with a diameter of approx. 7 mm and 0.5 mm thick, self-prepared bone punches made from pig shoulder blades (10 mm in diameter) which, in addition to the outer cortex (1–2 mm), also contained parts of the cancellous bone of different thicknesses. In addition, we made 10 × 10 mm specimens from human femoral neck (*Collum femoris*). The samples were used with the consent of the donors and in accordance with the guidelines of the Ethics Committee of the University Hospital Düsseldorf (study number 3634) and in compliance with the Declaration of Helsinki principles (revision of October 2013). With these human specimens, we were able to produce bone samples of various thicknesses by removing bone tissue using a medical circular bone saw.

### 2.7. CAP Treatment of Bone Samples

The cortical tissue chips were equilibrated in a buffered aqueous solution for 10 min before use. The bone chips were either treated individually with plasma or, in order to characterize the penetration depth of NO derivatives (NODs) into the bone samples, several such chips were stacked on top of one another and treated with the CAP from above. In order to prevent the lateral entry of plasma components into this bone stack, the stack was sealed with wax.

The native porcine bone punches, which in cross-section form a “sandwich structure” consisting of cancellous tissue enclosed upwards and downwards by cortical bone tissue, were exposed to CAPs on one side from above. Alternatively, this “sandwich structure” was cut in half across the cancellous tissue using a sharp scalpel or a saw and the specimens were treated with plasma from the cortical side. The 10 × 10 mm specimens of the human femoral neck were treated identically. The bone samples were treated with CAPs for 1, 5, 10, 15, 20 or 30 min in various experimental conditions described above. The power dissipated in the discharge was 122 to 410 mW, so that the dissipated electrical work was calculated at 0.122 to 12.3 W min. At the indicated time points after plasma exposure, the bone samples were transferred to 5 mL PBS.

In order to characterize the penetration depth of NODs into the bone tissue after CAP treatment (5 min, 410 mW power dissipated in the discharge), tissue layers were removed millimeter by millimeter from the apical side of the CAP-exposed bone samples using cutting tools and coarse files and the removed tissue layers were then each transferred to 2 mL of PBS.

The NODs contained in plasma-exposed bone tissue were washed out as completely as possible by gently shaking the PBS solution for 30 min, into which the bone preparations had previously been transferred. The NOD content of these aqueous exudates was quantified using the CLD as described above.

In order to detect and quantify a release of native nitric oxide from DBD plasma-treated (10 min 275 mW power dissipated in the discharge) human bone preparations of the femur neck (10 × 10 mm), these preparations were transferred to the gastight chamber immediately after plasma treatment and NO release was quantified by CLD as described above.

For all bone tissue samples, the NOD values obtained after DBD plasma exposure were normalized to the weight of the samples.

### 2.8. Bacterial Culture and Evaluation of the Bactericidal Effect of DBD Plasmas

(1)Test bacterium

*Staphylococcus epidermidis* stock solution with a McFarland value of 0.5 mcf contained a bacterial concentration of 1.5 × 10⁸ cfu/mL. The bacterial culture was carried out on 9 cm bacterial culture plates with Mueller Hinton 2 agar + 5% sheep blood (MHS, from bioMérieux SA, Marcy l’Etoile, France.

(2)Bactericidal impact of plasma treatment of sterile bone samples on the subsequently inoculated bacteria.

This experimental set-up was intended to check whether a CAP pretreatment of bone tissue can develop a bacteriostatic or bactericidal effect during a subsequent inoculation of the plasma treated samples. For this purpose, the bone samples were exposed to CAPs (10 min 275 mW power dissipated in the discharge) and approximately 20 min later, these bone samples were inoculated on the plasma exposed surface of the sample or alternatively on the opposite, non-plasma treated, sample side with 1.5 × 10^6^ bacteria. After 24 h of incubation in a moisture 37 °C temperate chamber, the infected samples were transferred to 600 mL of sterile PBS. In order to detach the bacteria, the bone samples were shaken intensively for 30 min with an orbital shaker (vortex) and then wiped off intensively with a cotton swab. Following this procedure, the tip of the cotton swab was cut off and taken up in the existing sample volume and vortexed for a further 5 min. Subsequently, 200 μL of this bacterial solution was plated in different dilutions to agar plates and after a further 24 h, the number of visible bacterial colonies was documented and the reduction in the bacterial count (log_10_ reduction of cfu/mL) was calculated. Bone samples inoculated with bacteria but not treated with plasma served as positive controls. As a negative control, we used plasma treated bone samples that were not inoculated with bacteria.

(3)Bactericidal impact of plasma treatment of bone samples inoculated with bacteria.

With this second experimental setup, we wanted to evaluate the effect of DBD CAPs on bone samples that were already inoculated by bacteria. For this purpose, the bone preparations were inoculated with 1.5 × 10^6^ bacteria and the bacterially infected surface was treated with the DBD plasma (10 min 275 mW power dissipated in the discharge), either 20 min or 24 h after the bacterial inoculation. In order to record bactericidal in-depth effects of the plasma, we treated, in alternative test approaches, the non-infected sample side with plasma. After 24 h since the last plasma exposure, the samples were transferred to 600 mL of PBS and the reduction in the bacterial count (log_10_ reduction of cfu/mL) was calculated identically as described above in the last chapter. Again, bone samples inoculated with bacteria, but not treated with plasma, served as positive controls, whereas sterile bone samples that were otherwise treated identically served as negative controls.

(4)Characterization of the bactericidal effect of plasma pretreatment of agar plates followed by bacterial inoculation.

In order to further elaborate the therapeutic relevance of plasma pretreatment of a tissue as a preventive and protective option against bacterial infections, we treated selected areas of agar plates for 5 to 30 min with DBD CAPs (275 mW power dissipated in the discharge). After 24 h, we inoculated the plates with bacteria and characterized the bacterial growth after a further 24 h of incubation.

### 2.9. Statistical Analysis

For statistical analysis, we used GraphPad Prism 8 (San Diego, CA, USA). Significant differences were evaluated using either paired two-tailed Student’s *t*-test or ANOVA followed by an appropriate post hoc multiple comparison test (Tukey method). A *p*-value < 0.05 was considered significant.

## 3. Results

### 3.1. Exposure Time- and Performance-Dependent Accumulation of NO Derivatives into CAP-Exposed Aqueous Solutions and Bone Samples

CAP treatment of the aqueous solution led to an exponential increase in the accumulated amounts of nitrite, which was dependent on the exposure time and electrical work dissipated in the discharge (Figure 1A). In parallel, we observed a decrease in the solution pH, which aimed for a limit value of approx. pH 3.5 (Figure 1B). The increase in nitrite accumulation and the corresponding decrease in the solution pH correlated positively with a significant generation of gaseous NO from plasma exposed samples, as quantified by CLD (Figure 1C).

Additionally, we treated bone punch preparations from the pigs shoulder with DBD plasmas of different power dissipated in the discharge. As shown in Figure 2A–C, bone tissue can be “loaded” with NODs. The amount of accumulated nitrite correlated positively with exposure time and the increasing power dissipated in the discharge. As part of the differential analysis of the accumulated NOD species, we found with human bone preparations that the majority of the NODs represent nitrate (Figure 2E), here detected in mM amounts, and about 1/20th to 1/50th of the NOD was stored in the form of nitrite (Figure 2D).

### 3.2. Quantification of NO Emanation from CAP-Exposed Bone Samples

In order to quantify NO release from plasma-treated bone samples, we transferred the samples after the corresponding plasma exposure into a gas-tight chamber which was filled with an inert carrier gas (N_2_) (Figure 3A). After every two minutes, we quantified the accumulated NO amount by CLD. We observed a significant release of NO from human bone samples exposed to DBD plasma. The amount of NO released from the bone tissue (500–800 mg) was in the range between 15 and 22 pmol/min, immediately after the plasma exposure (Figure 3B). In the further course, the NO emanation from the bone tissue constantly decreased, but eight hours after the plasma treatment, it still averaged approx. 10 pmol/min. In parallel, we observed a significant decrease in the pH of DBD plasma exposed bone sample (Figure 3D).

### 3.3. Characterization of the Penetration Depth and Distribution of NO Derivatives in Bone Tissue Exposed to DBD Plasma

Knowledge of the depth of penetration of NODs into the treated bone tissue is of great relevance in a possible plasma-based therapy. We, therefore, treated native human bone samples of the femur neck with DBD plasma (5 min 410 mW) and quantified the concentration of nitrite at different depths and time points after the plasma treatment (Figure 4). In the depth intervals studied and under the conditions described, we could observe a high accumulation of nitrite in the upper layers of the treated bones, 30 min after plasma exposure. However, with an increasing depth of the bone tissue, the nitrite concentration decreased sharply. Interestingly, 90 and 180 min after the plasma treatment, we were able to detect a steadily increasing nitrite concentration also in the deeper regions of the plasma-exposed bone, which indicates a diffusion-controlled distribution of the NODs transferred into the bone tissue (Figure 4).

### 3.4. Bactericidal Impact of Plasma Treatment of Bone Samples Inoculated with Bacteria

In order to be able to better assess the bactericidal effect of plasma treatment on bone tissue, we have inoculated the bone preparations (punch preparations from the pork shoulder blade) with a *Staphylococcus epidermidis* strain (1.5 × 10^6^). After 24 h, we treated the bacterially inoculated side (Figure 5A) or the non-inoculated opposite side of the specimens (Figure 5B) with CAPs (power dissipated in the discharge was 275 mW) for 10 min. After a further 24 h, we dissolved the bacteria from the plasma-treated samples and the untreated bone preparations, applied the resulting bacterial solutions in different dilutions to agar plates, and determined the reduction in the bacterial count (log10 reduction of cfu/mL). As shown in Figure 5A, plasma treatment of the bacterially inoculated side led to a log_10_ 3.92 ± 0.13 reduction and of the non-inoculated opposite side of the specimens, to a log_10_ 3.73 ± 0.27 reduction in cfu/mL (Figure 5B).

In an alternative experimental setup, the results of which are shown in Figure 6, we treated bacterially inoculated bone preparations with DBD plasma identically as described above, but immediately, about 10 min after inoculation. Plasma treatment of the bacterially inoculated side (Figure 6A) led to a log_10_ 3.76 ± 0.34 reduction and of the non-inoculated opposite side of the specimens to a log_10_ 3.71 ± 0.30 reduction in cfu/mL (Figure 6B).

### 3.5. Bactericidal Effect of Plasma Pretreatment of Agar Plates Followed by Bacterial Inoculation

In order to check to what extent plasma exposure could also serve as a preventive or prophylactic bactericidal therapy option, we treated agar plates with CAPs generated with the power dissipated in the discharge of 122 mW, 221 mW or 275 mW (**C**), for 5 min, 10 min, 20 min or 30 min. After 24 h, agar plates were inoculated with the bacteria. As shown in Figure 7, bacterial growth was apparently inhibited on the plasma exposed agar areas (Figure 7A–C). The growth inhibition of bacteria apparently correlated with the exposure time and the power dissipated in the discharge (Figure 7D).

### 3.6. Bactericidal Impact of Plasma Treatment of Sterile Bone Samples on the Subsequently Inoculated Bacteria

In a modified experiment with punch preparations from the pork shoulder blade, we treated the bone samples from one side with the CAPs (10 min 275 mW) and after 24 h we inoculated either the treated or the untreated side of the bone sample with bacteria. After a further 24 h, we determined the reduction in the bacterial count (log10 reduction of cfu/mL) as described above. In Figure 8, we show that bacterial inoculation on the plasma treated side (Figure 8A) led to a log_10_ 3.68 ± 0.26 reduction and of the non-inoculated opposite side of the specimens, to a log_10_ 3.46 ± 0.27 reduction in cfu/mL (Figure 8B).

## 4. Discussion

Bacterial infections of the bone tissue, such as osteitis or osteomyelitis, are very serious and unfortunately not uncommon complications and major challenges in trauma and orthopedic surgery. Larger orthopedic interventions or fractures with severe tissue injuries are often associated with a disturbed local immune response, which promotes implant-associated infections due to opportunistic pathogens. A total of 60% of all implant-associated infections are caused almost equally by two strains of staphylococci, *S. aureus* (34%) and *S. epidermidis* (32%) [26,27]. Microorganisms that have penetrated the wound can then settle on the surface of implants and devitalize areas of bone tissue, where they form a biofilm, a hydrated matrix of extracellular components. In the biofilm, dividing bacteria can be found, which in this planktonic form are potentially sensitive to antibiotics, but most of the bacteria in the biofilm are in their sessile form, which is characterized by a reduced metabolism and reproduction rate, and are therefore far less sensitive to bactericides [28]. The free-swimming planktonic form of bacteria are mainly found in the expansion phase necessary for the colonization of new habitats. Compared to the planktonic form, the biofilm variant offers up to four orders of magnitude higher resistance and tolerance to the immune system, biocides and antibiotics [29,30]. This fact shows how difficult it can be to eradicate biofilm infections. Additionally, the matrix of the biofilm represents an effective barrier between bacteria and immunocompetent cells of the innate or acquired immune response, as well as an effective barrier against antibiotics [31].

The biofilm is the predominant form of bacterial growth on implants and the driving pathological source of chronic ostitis [32,33]. Almost all implant-associated infections caused by biofilm-forming bacteria lead to the development of subacute to chronic ostitis, which is accompanied by progressive chronic bone degeneration with impaired and delayed healing. In order to avoid life-threatening complications, such as sepsis, adequate therapy with an attempt to completely eliminate the pathogen causing ostitis is necessary [34]. Radical debridement, lavage, and removal of implants in combination with systemic or oral antibiotic therapy are, therefore, the gold standard of treatment [35]. However, it should not go unmentioned here that intravenous or oral antibiotic therapies can lead to severe systemic side effects, and often only have minor effects due to insufficient antibiotic concentrations at the local infection site [36].

To date, some molecular factors have been identified which are able to induce a transformation of the antibiotic-resistant sessile and surface-associated biofilm phenotype to the antibiotic-susceptible planktonically spreading phenotype in a biofilm. One of these factors is NO [19,20]. Even very low NO concentrations in the nano molar range lead to an increased dispersion of the biofilm and at the same time, to an increased sensitivity of the biofilm and the dispersed bacteria to several classes of antimicrobial agents, as well as a strong increase in the effectiveness of these antibiotics [37,38]. The dispersion properties of NO seem to be well conserved across bacterial species [19,20]. The ability of NO to increase the dispersion of biofilms represents a great opportunity for the development of new and more efficient therapeutics to control biofilm-related infections and to overcome biofilm resistance. The bacteria would not be directly killed by the low doses of NO, but the released planktonic cells would have a decidedly increased susceptibility to antibiotics and other antimicrobial agents and could thus be effectively eliminated [38,39]. Moreover, NO-based strategies to reduce or eliminate biofilms would greatly benefit from combined treatments with standard antibiotics to prevent or eliminate bacterial infections. In the case of higher NO concentrations, the development of further and significantly more reactive and pathogen-killing nitrogen species (RNS) would be favored, whereby a damaging effect of these RNSs on host cells would also be expected [40]. RNSs can significantly change the functionality of nucleic acids and proteins through nitrosylation or nitration of amine, thiol and tyrosine residues as well as metal centers and, under certain circumstances, irreversibly damage it [41]. Due to the good diffusion capacity of RNS and the aforementioned diverse modes of action, RNSs are effective broad spectrum antimicrobial agents at higher concentrations, which could kill biofilms of gram-negative and gram-positive bacteria [19,20].

At this point, we would like to point out that we here deliberately omit the entire topic of the role, regulation and support of local enzymatic NO production and focus exclusively on therapy options addressed by exogenous application of NO. Interestingly, regarding the clinical picture of ostitis, we could not find any evidence of experimental or clinical therapeutical use of NO in the current literature databases. The reason for this is undoubtedly the lack of availability of suitable NO sources that could be used therapeutically meaningfully after removing the implants, debridement and lavage. The two essential exogenous NO sources that have already been successfully used, e.g., in the treatment of bacterially infected wounds in vivo, are gaseous NO and NO donors [42]. Both are, however, for the local treatment of osteitis or osteomyelitis unsuitable. The long therapeutic duration of use of gaseous NO in high concentrations represents a considerable source of danger and stress for the health of all parties, regardless of the high costs, and therefore is clinically impractical. The use of NO donors for pharmacological therapy of an already existing ostitis appears unsuitable as well, as the degradation of these substances, i.e., the release of NO, varies greatly depending on the humidity, the pH value and the temperature of the environment [43]. Therefore, with NO donors, the desired release kinetics and effective NO concentration cannot be specifically estimated or predicted. It should not go unmentioned, however, that materials scientists in particular, when functionalizing implants with spontaneously NO-releasing systems, pursue the idea of counteracting bacterial infection of the bone as a preventive measure [44].

In contrast to the two NO sources mentioned, a NOD-plasma-based therapeutic approach would be technically and pharmacologically meaningful. By using flexible DBD electrode mats, which can adapt to the anatomical topography of a bone in comparably large areas, treatment could occur in one step. Such a technology, adapted to DBD technology, is already available and could easily be optimized for the situation of an exposed infected bone. In addition, NOD-accumulating CAPs generated by other plasma technologies could of course also be used, such as the indirectly generated CAPs in the plasma-jet variant.

With the help of NOD-plasma-application, a multitude of therapy-specific goals on non-infected, as well as already bacterially infected bone tissue, could be addressed by combining the duration of application and the power dissipated in the discharge. In the case of non-infected bones, as a preventive treatment, particularly endangered areas could be treated differently depending on the existing risk of bacterial infection. We are thinking, for example, of the areas of the bone fracture after the bone has been repositioned and retained. In the case of open fractures and environmental exposure of the bone stumps, a preventive NOD-plasma-exposure of the bone stumps prior to repositioning could help to prevent infections. In addition, preventive treatment of bone areas that are particularly susceptible to infection would be justified. Of particular importance here are the screw holes, which represent a relevant source of risk for pathogens to penetrate deeper areas of the bone tissue. In each of the scenarios mentioned, the NOD-plasma-exposed bone tissue could be “charged” to different degrees with acidified NO derivatives, depending on the duration of exposure and discharge capacity, analogous to our results that we carried out with human skin tissue [15]. The resulting release of NO in an acidic environment, alone or synergistically with antibiotics, would help to prevent a possible bacterial contamination and infection of the operating area during a surgical procedure.

At the time of our investigations with human skin tissue [15], however, it was not yet clear whether the observed accumulation of NODs in the plasma-exposed samples was directly due to the NO contained in the plasma, or was mediated by nitrogen oxides other than NO. The data presented here strongly suggest that the NO contained in the plasma is not primarily responsible for the effects observed in our bone model. The DBD source used here generated, at the lowest discharge power (125 mW), plasmas with a NO concentration of approximately 200 ppb, but an approximately 500-fold higher concentration of NO_2_ (100 ppm) [15]. We therefore postulate that the NO_2_ contained in the CAP in the moist environment of the exposed biological tissue essentially leads to the generation of nitrous acid (HNO_2_) and nitric acid (HNO_3_) (Equation (1)). Nitrous acid (HNO_2_) generates consequently gaseous NO (Equation (2), Equation (3)), as well as the formation of nitrite (Equation (4)) that under acidic conditions supports HNO_2_ formation and therefore additional release of NO (Equation (5)).
2NO_2_ + H_2_O ⇌ HNO_3_ + HNO_2_(1)
3HNO_2_ ⇌ HNO_3_ +2NO(g) + H_2_O(2)
HNO_2_ ⇌ NO + NO_2_ + H_2_O(3)
2NO_2_(g) + H_2_O ⇌ 2H^+^ + NO_2_^−^ + NO_3_^−^.(4)
H^+^ + NO_2_^−^ ⇌ HNO_2_(aq) HNO_2_(g)(5)

These assumptions are supported by our observations that the NOD-plasma treatment of the aqueous solutions led to a significant pH decrease and aimed for the pH value of 3.29 that is characteristic of HNO_2_, and at the same time induced a long-lasting release of NO from the exposed solution.

In the concept of the NOD-plasma-based therapy option against biofilms, however, another aspect would play the essential role. Under acidic conditions, nitrite reacts to form nitrous acid and other nitrogenous metabolites [45] and shows a pronounced antimicrobial effect against bacteria, fungi and the common oral and skin pathogens, as well as bacterial spores [46,47]. Interestingly, acidified nitrite also enhances the antibacterial and fungicidal effects of hydrogen peroxide [48]. In this context, it is important to mention that DBD CAPs, in addition to NODs, also lead to accumulation of a not inconsiderable amount of hydrogen peroxide into treated biological samples [49]. It is therefore very likely that these two factors are largely responsible for the antimicrobial effect of CAPs, as was already postulated by Naitali et al. [50]. In particular, the generation of peroxynitrite from the reaction of nitrite and H_2_O_2_ under acidic conditions could be the reason behind the increased antimicrobial effect of CAPs [51].

Of course, serious concerns about possible risks and side effects could be raised with regard to CAP-based therapy on bone tissue. With intensive use of NOD-plasma, accompanied by a temporary strong acidification of the bone tissue and the entry of high amounts of NODs, as would be the case in the treatment of severe biofilm contamination of bone tissue, cytotoxic events on cells of the bone tissue could not be completely avoided. Previous data show [52,53] that ROS/RNS-containing plasmas can have a pronounced cytotoxic character, depending on the duration of application or ROS/RNS concentration. The cytotoxic mechanism includes a structural change in proteins, nucleic acids and lipid membranes caused by oxidative and nitrosative stress leading to the loss of the biological function of these structures, up to the induction of apoptotic or necrotic cell death. It is also such a cytotoxic aspect of CAPs that could limit potential therapeutic applications based on the intensive use of highly reactive plasmas. However, the cytotoxic character of potentially cytotoxic plasmas in the treatment of bacterially infected bone tissue would occupy a special position. In such a case, even a partial devitalization of the treated bone tissue would be tolerable to a large extent, as it can be assumed that such a devitalization would probably only be a temporary side effect of plasma therapy. As we show here, even with the highest rate of accumulation of NO derivatives, we could no longer detect any relevant amounts of NO activity in bone tissue, at the latest 72 h after exposure to plasma. This would enable the remaining sterile bone matrix tissue during the subsequent post-therapy regeneration phase to be repopulated by cells immigrating from the biologically intact edges and a revitalized intact bone tissue would be created again.

## 5. Conclusions

DBD-generated CAPs, energetically optimized for nitric oxide chemistry, can “charge” bone tissue with appropriate amounts of NO derivatives and lead to the establishment of a therapeutically relevant NO-releasing system in bone tissue. The quantity of NO equivalents introduced into the bone tissue appears to be solely a function of the exposure time and the amount of electrical work used to generate the plasma. It is thus possible, depending on the therapeutic needs required, to treat bone tissue in a targeted manner on its surface or also in the depth of the tissue with therapeutically relevant amounts of NOD. In the near future, depending on the amount of enriched NODs in the bone tissue, NOD-plasma-assisted therapy of bacterial osteitis could achieve both NO-dependent spread and NOD-induced destruction of a biofilm. Thus, this technology could be used successfully as an accompanying therapy option to other forms of therapy, but also as the sole therapy option in the fight against bacterial bone infections.

## Figures and Tables

**Figure 1 biomedicines-11-00244-f001:**
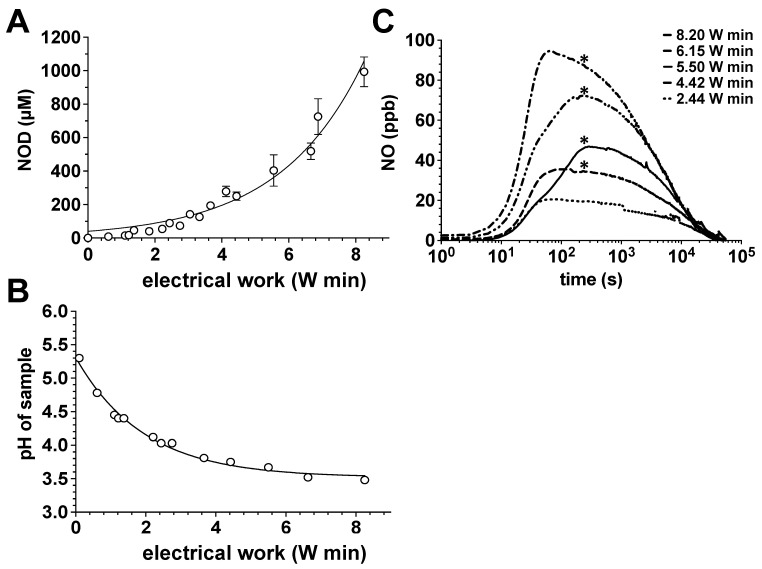
Impact of the power dissipation in the discharge on the pH, NOD accumulation and NO release in aqueous solutions exposed to the CAPs. Using the DBD plasma source, 20 mL of water samples were treated with CAPs generated with the power dissipated in the discharge of 122, 221, 275 or 410 mW for 5, 10, 15 or 20 min. The nitrite accumulation in the treated solutions was then quantified by CLD, the pH value of the solution was assessed with a pH meter and the course and the amount of NO release from these solutions was also characterized by CLD. (**A**) Nitrite content of the plasma-treated samples as a function of the electrical work dissipated in the discharge. Open circles represent the mean ± S.D. of four individual experiments. (**B**) pH values of the plasma-treated samples as a function of the electrical work dissipated in the discharge. (**C**) NO release from samples treated with plasma. A representative result of six individual tests is shown. *, *p* < 0.05, as compared to NO release from solutions treated with plasmas generated under conditions of less electrical work dissipated in the discharge.

**Figure 2 biomedicines-11-00244-f002:**
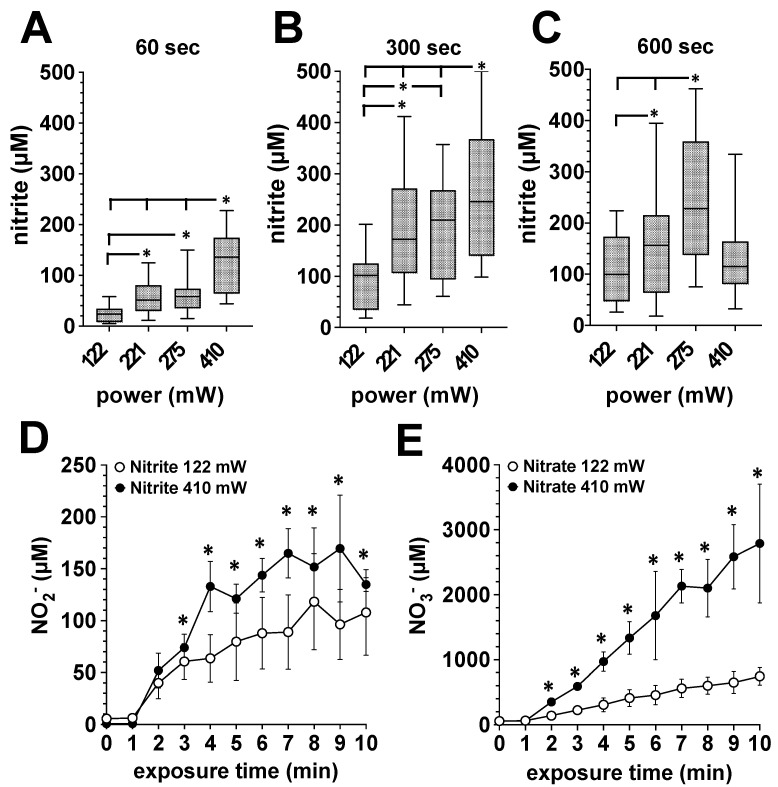
Influence of the power dissipation in the discharge on the plasma-induced accumulation of nitrite and nitrate in exposed bone tissue. Using the DBD plasma source, bone tissue samples were treated with plasma generated with the power dissipated in the discharge of 122, 221, 275 or 410 mW for for 1, 5 or 10 min. The nitrite and nitrate accumulation in the bone tissue was quantified by CLD, as described in the Material and Methods section. **A**–**C**, Porcine bone tissue treated 60 s (**A**), 300 s (**B**) or 600 s (**C**) with plasma generated with the power dissipated in the discharge of 122, 221, 275 or 410 mW. Values of 9 individual (*n* = 9) experiments are shown as boxplot with median value and with whiskers with minimum and maximum. *, *p* < 0.05. (**D**), Accumulation of nitrite or nitrate (**E**) in human bone tissue from the femoral head shaft after exposure to plasma generated with the power dissipated in the discharge of 122 mW (white circles) or 410 mW (black circles). Values represent the mean ± S.D. of four individual experiments. *, *p* < 0.05 as compared to the corresponding values obtained with the power dissipated in the discharge of 122 mW (white circles).

**Figure 3 biomedicines-11-00244-f003:**
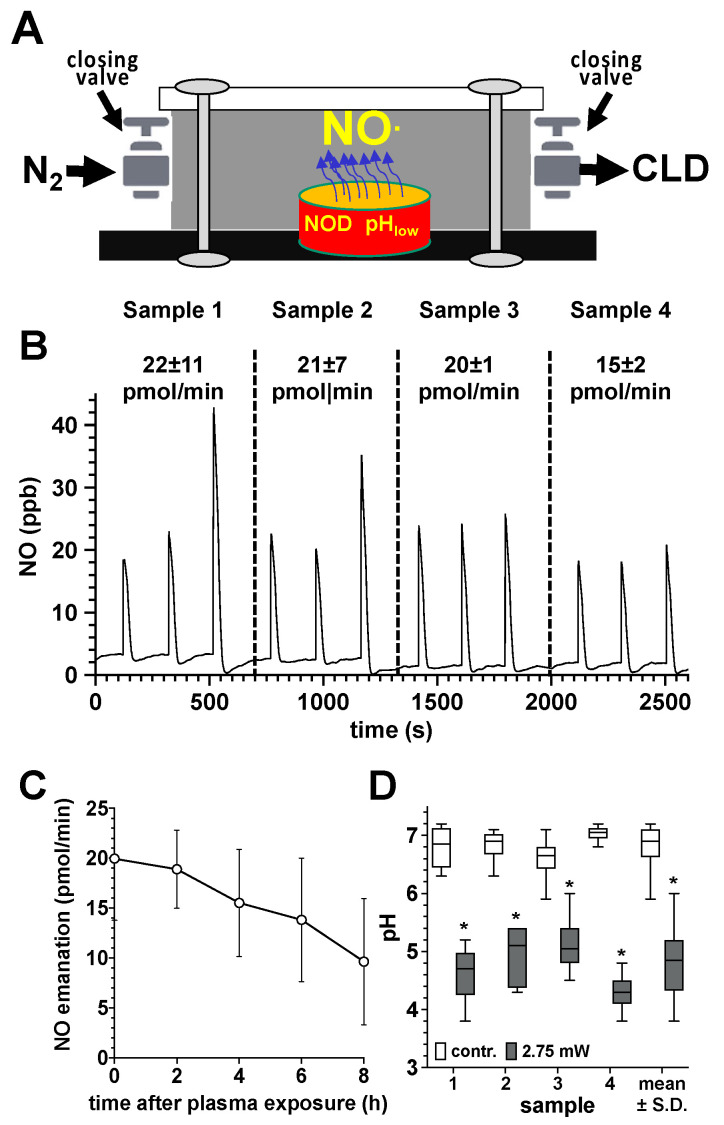
Quantification of NO release from plasma-treated human bone samples. (**A**), sketch of the experimental setup for quantifying the NO release of plasma-exposed bone tissue. DBD plasma-treated human bone samples (10 min at 275 mW power dissipation in the discharge) of the femur neck (500–800 mg) were transferred to a hermetically sealed chamber (35.2 cm^3^), flushed with nitrogen to remove oxygen immediately after the plasma exposure. NO liberated from the bone samples was accumulated in the chamber for 120 s and the NO content was quantified by flushing the gas contents of the chamber into the CLD analyzer. (**B**), NO release (pmol/min) from the treated bone tissue (mean values ± S.D. of three measurements (triplicates) of each of the four samples (samples 1–4). (**C**), Characterization of the NO emanation (pmol/min) from the tissue of the four plasma-treated bone samples at different time points after the plasma treatment. (**D**), pH values of the treated surface of the four plasma treated bone samples immediately after plasma exposure, as detected by using a flat membrane pH electrode. *, *p* < 0.05 as compared to the control samples.

**Figure 4 biomedicines-11-00244-f004:**
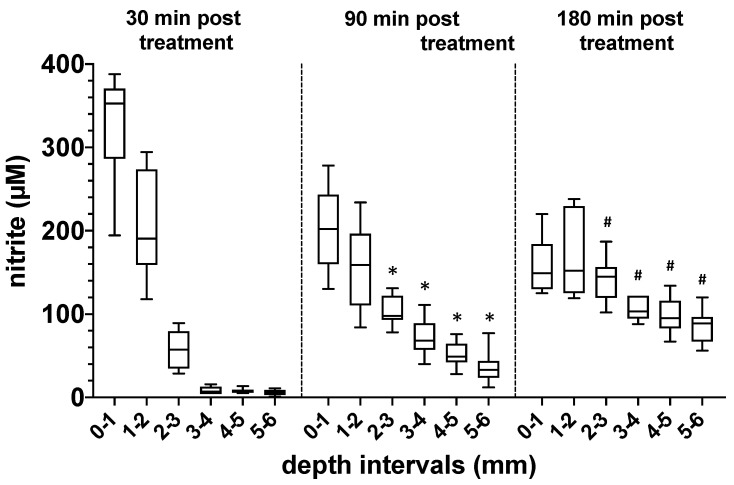
Evaluation of the depth of penetration of nitrogen oxide derivatives into bone tissue after exposure to plasma. Quantification of the nitrite content (µM) at different depths of human bone samples treated with DBD plasma (5 min at 410 mW power dissipation in the discharge). Values from 5 individual (*n* = 5) experiments are presented as a boxplot with median and with whiskers with minimum and maximum. The nitrite content in the bone samples was detected 20, 90 and 180 min after plasma exposure using CLD technology. *, *p* < 0.05 as compared to corresponding values detected after 20 min. ^#^, *p* < 0.05 as compared to corresponding values detected at 20 or 90 min.

**Figure 5 biomedicines-11-00244-f005:**
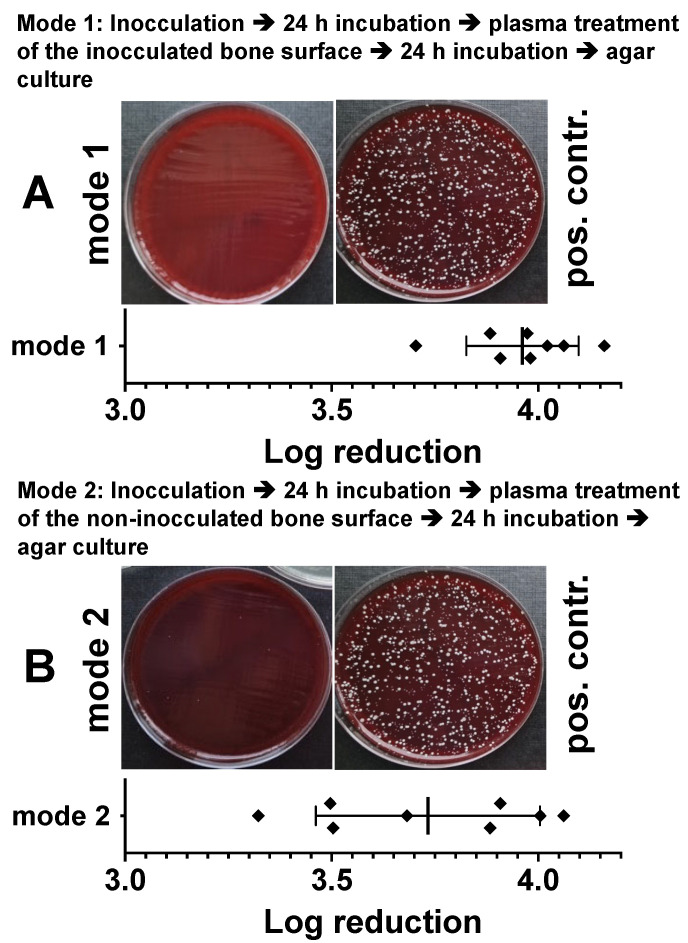
Characterization of the antibacterial effect of DBD plasma in a model of a bacterially infected bone preparation. Punch preparations from the pork shoulder blade were inoculated with the *Staphylococcus epidermidis* strain (1.5 × 10^6^). After 24 h of incubation, the bacterially infected area (**A**, mode 1) or the non-infected side (**B**, mode 2) of the specimens were exposed to the DBD CAPs (5 min at 410 mW power dissipation in the discharge). After a further 24 h, the bacteria were detached and the reduction in the bacterial count (log_10_ reduction of cfu/mL) was calculated. As positive controls (pos. contr.), the corresponding bone samples were treated identically with the bacteria, but were not exposed to plasma. Bars shown represent the mean ± S.D. of six individual experiments.

**Figure 6 biomedicines-11-00244-f006:**
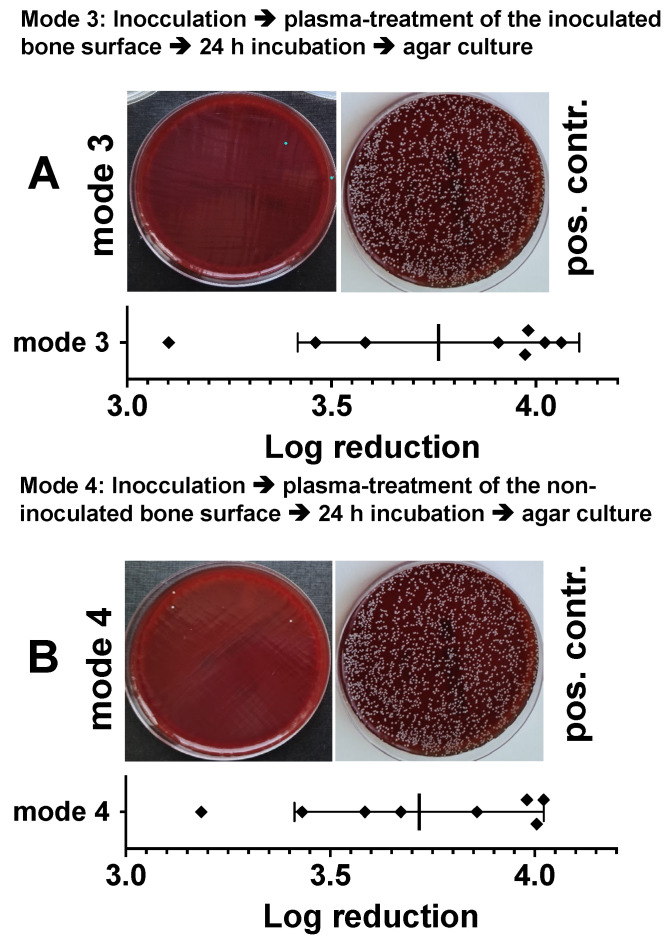
Characterization of the antibacterial effect of DBD CAPs in a model of a bacterially contaminated bone preparation. Punch preparations from the pork shoulder blade were inoculated with the Staphylococcus epidermidis strain (1.5 × 10^6^). Immediately (approximately 10–15 min) after bacterial inoculation, the non-contaminated area (**A**, mode 3) or the bacterially contaminated side (**B**, mode 4) of the specimens were exposed to the DBD plasma (10 min at 275 mW power dissipation in the discharge). After 24 h, the bacteria were detached from the bone preparations and the resulting bacterial solutions of mode 3 or mode 4 were spread out on agar plates (photographic images of mode 3 or mode 4). As a positive control (pos. contr.), the corresponding bone samples were treated identically, but not exposed to plasma. The quantitative evaluation was carried out by counting the photographically recorded colonies on the agar plate using the *imageJ* software. Bars shown represent the mean ± S.D. of six individual experiments. *, *p* < 0.05 as compared to the positive control (pos. contr.).

**Figure 7 biomedicines-11-00244-f007:**
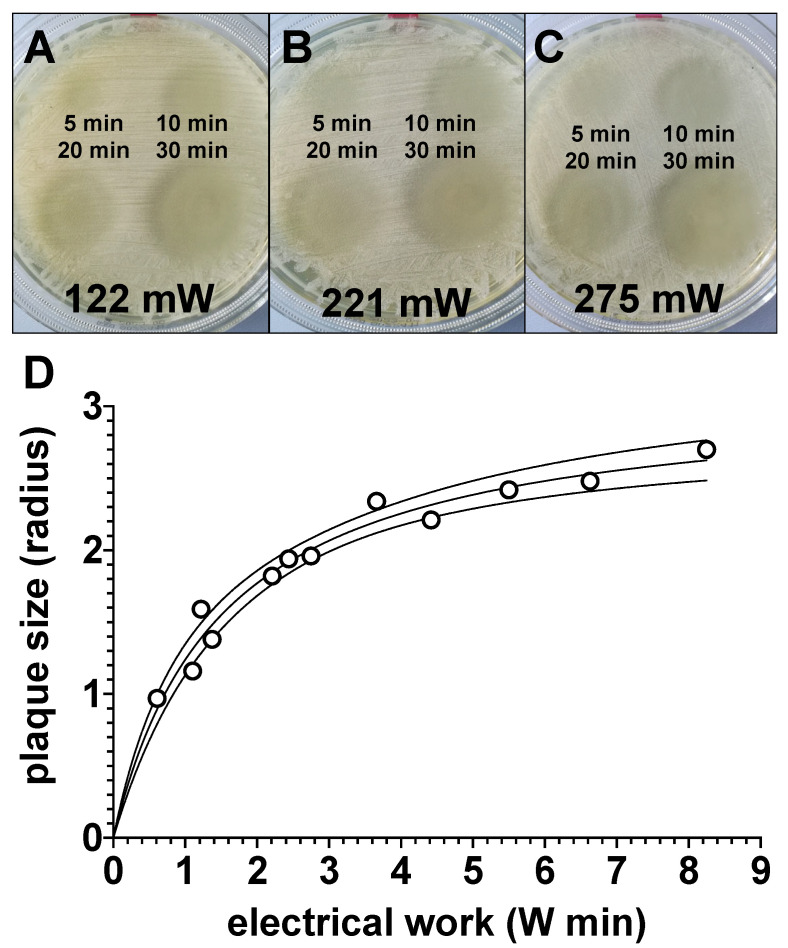
Evaluation of the effect of preventive plasma treatment of bacterial growth agar plates prior to bacterial inoculation. Four quadrants of bacterial agar growth plates were treated with plasma, generated with the power dissipated in the discharge of 122 mW (**A**), 221 mW (**B**) or 275 mW (**C**), for 5 min (top left), 10 min (top right) 20 min (bottom left) or 30 min (bottom right). The used electrode had a total diameter of 10 mm. A total of 24 h after plasma exposure, agar plates were inoculated with bacteria. After a further 24 h, the radii of the bacteria-free areas, as shown in (**A**–**C**), were measured with a ruler and these values were plotted graphically in relation to the electrical work dissipated in the discharge (**D**). Values shown in (**D**) represent the mean of two individual experiments. The solid line represents the interpolations line; the area between the solid lines represents 95%-asymptotic confidence interval.

**Figure 8 biomedicines-11-00244-f008:**
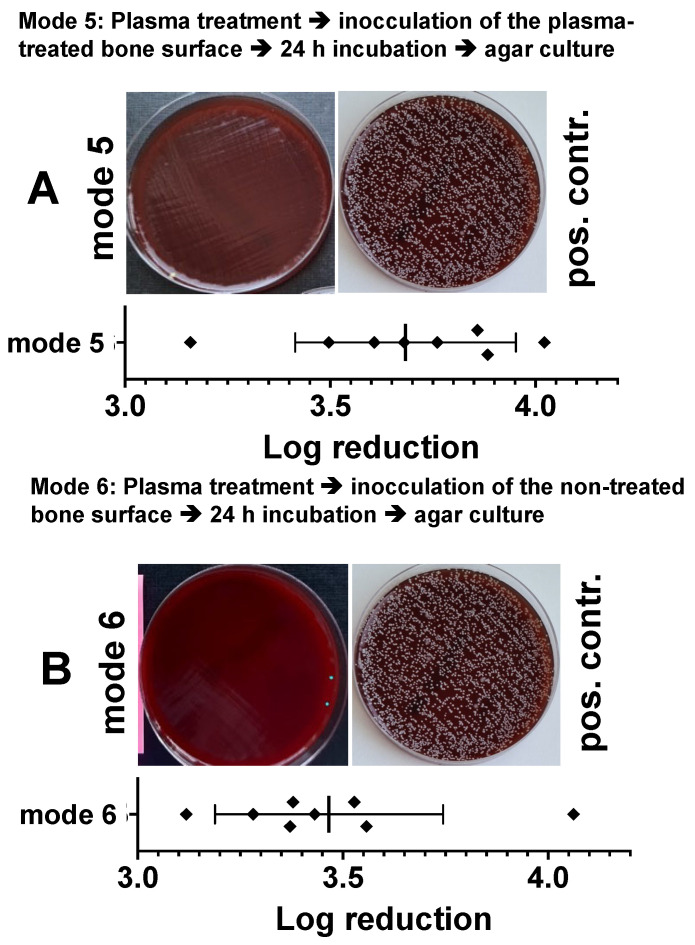
Characterization of the antibacterial effect of a preventive DBD CAP treatment in a model of a bacterially infected bone samples. Punch preparations from the pork shoulder blade were exposed to the DBD plasma (10 min at 275 mW power dissipation in the discharge). Ten minutes after plasma exposure, the non-treated side of the sample (**A**, mode 5) or the plasma exposed bone area of the samples (**B**, mode 6) were inoculated with the Staphylococcus epidermidis strain (1.5 × 10^6^). After 24 h, the bacteria were detached from the bone preparations and the resulting bacterial solutions of mode 5 or mode 6 were spread out on agar plates (photographic images of A; mode 5 or B; mode 6). As a positive control (pos. contr.), the corresponding bone samples were treated identically, but not exposed to plasma. The quantitative evaluation was carried out by counting the photographically recorded colonies on the agar plate using the *imageJ* software. Bars shown represent the mean ± S.D. of six individual experiments. *, *p* < 0.05 as compared to the positive control (pos. contr.).

## Data Availability

The datasets used and/or analyzed during the current study are available from the corresponding author on reasonable request.

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
