# Peer review of "Enrichment of Bone Tissue with Antibacterially Effective Amounts of Nitric Oxide Derivatives by Treatment with Dielectric Barrier Discharge Plasmas Optimized for Nitrogen Oxide Chemistry"

_biomedicines, 2023, doi:10.3390/biomedicines11020244_

Round 1
Reviewer 1 Report
This paper presents on Enrichment of bone tissue with antibacterially effective amounts of nitric oxide derivatives by treatment with dielectric barrier discharge plasmas optimized for nitrogen oxide chemistry. In this paper, they have described the current plasma technology opens up effective NO-based therapy options in the treatment of local bacterial infections of the bone tissue through the possibility of a targeted modulation of the NOD content in the generated CAPs. However, there are a few points that need to be clarified.
1. Authors mentioned ROS-mediated bone regeneration, however, the exact mechanism leading to tissue regeneration is not clearly mentioned.
2. Authors performed experiments related to Invitro ROS levels.
3. The title of the study seems not suitable. Please modify the title so that it appropriately matches with the target study.
4. Authors should add More references concerning the materials should be cited to tell the readers the position of this work to contribute to this research field: ACS Bio Materials, 2020, 3, 2218-2229., ACS Appl. Mater. Inter., 11 (2019) 288-299., Biomaterials, 288, 2022, 12172, etc.
Author Response
Reviewer#1
1.) Authors mentioned ROS-mediated bone regeneration, however, the exact mechanism leading to tissue regeneration is not clearly mentioned.
We are very sorry, but in our manuscript, we could not find any passage in which we treat the topic "ROS-Mediated Bone Regeneration".
2.) Authors performed experiments related to Invitro ROS levels.
Yes, that's right. We assume that the oxidative or nitrisative stress, which is generated by the DBD plasma in the bone tissue, is many times higher than the physiologically existing one. However, with the plasma we used, we did not want to address any physiolgic regenerative processes, but primarily achieve bacteria-toxic effects.
3.) The title of the study seems not suitable. Please modify the title so that it appropriately matches with the target study.
In the current study, we technically modify the DBD plasma source used in such a way that we achieve an optimal NO chemistry in the generated plasma. With these plasms we treat bone tissue and introduce NO derivatives into the bone tissue and check to what extent we can achieve an antibateral effect with this application mode. Against this background, the chosen title “Enrichment of bone tissue with antibacterially effective amounts of nitric oxide derivatives by treatment with dielectric barrier discharge plasmas optimized for nitrogen oxide chemistry” seems very suitable to us. If the reviewow could suggest a more suitable title, we would join this proposal.
4.) Authors should add More references concerning the materials should be cited to tell the readers the position of this work to contribute to this research field: ACS Bio Materials, 2020, 3, 2218-2229., ACS Appl. Mater. Inter., 11 (2019) 288-299., Biomaterials, 288, 2022, 12172, etc.
Apparently, something went wrong with the text formatting. Unfortunately, we cannot understand which change requests the reviewer intended. We will carry out evalual revision requests from the reviewer as part of the second review process.
Reviewer 2 Report
The paper entitled "Enrichment of bone tissue with antibacterially effective amounts of nitric oxide derivatives by treatment with dielectric barrier discharge plasmas optimized for nitrogen oxide chemistry" by Dennis Feibel and co, presents the possibility of DBD CAP-induced NOD accumulation and therapeutically relevant NO release in calcified bone tissue.
The subject is of great interest, but some minor revisions must be done, as follows:
1) The abstract is too long, please present only the main achievement of the paper
2) In my opinion, the possible risks and side effects (cytotoxic events on cell) of using NOD-plasma (page 17, row 570-582) must be discussed in more details
3) A Conclusion part is missing
Author Response
Reviewer#2
1.) The abstract is too long, please present only the main achievement of the paper
We have now sensibly shortened the abstract to less than 200 words.
2.) In my opinion, the possible risks and side effects (cytotoxic events on cell) of using NOD-plasma (page 17, row 570-582) must be discussed in more details
We have included the important aspect of possible side effects mentioned by the reviewer in the following text section (page 17, row 586 and following) and supplemented it with further information.
3.) A Conclusion part is missing
We've now slightly unformatted the last paragraph of the discussion and listed it as a Conclusion.